# N-type fast inactivation of a eukaryotic voltage-gated sodium channel

Jiangtao Zhang [1,2,9], Yiqiang Shi[3,9], Junping Fan[4,9], Huiwen Chen[5,6,9], Zhanyi Xia[2], Bo Huang [7], Juquan Jiang [6], Jianke Gong[1✉], Zhuo Huang [3✉] & Daohua Jiang [2,8✉]

Voltage-gated sodium (Na$_V$) channels initiate action potentials. Fast inactivation of Na$_V$ channels, mediated by an Ile-Phe-Met motif, is crucial for preventing hyperexcitability and regulating firing frequency. Here we present cryo-electron microscopy structure of Na$_V$Eh from the coccolithophore *Emiliania huxleyi*, which reveals an unexpected molecular gating mechanism for Na$_V$ channel fast inactivation independent of the Ile-Phe-Met motif. An N-terminal helix of Na$_V$Eh plugs into the open activation gate and blocks it. The binding pose of the helix is stabilized by multiple electrostatic interactions. Deletion of the helix or mutations blocking the electrostatic interactions completely abolished the fast inactivation. These strong interactions enable rapid inactivation, but also delay recovery from fast inactivation, which is ~160-fold slower than human Na$_V$ channels. Together, our results provide mechanistic insights into fast inactivation of Na$_V$Eh that fundamentally differs from the conventional local allosteric inhibition, revealing both surprising structural diversity and functional conservation of ion channel inactivation.

[1] College of Life Science and Technology, Key Laboratory of Molecular Biophysics of MOE, Huazhong University of Science and Technology, Wuhan, Hubei, China. [2] Laboratory of Soft Matter Physics, Institute of Physics, Chinese Academy of Sciences, Beijing 100190, China. [3] State Key Laboratory of Natural and Biomimetic Drugs, Department of Molecular and Cellular Pharmacology, School of Pharmaceutical Sciences, Peking University Health Science Center, Beijing 100191, China. [4] Beijing National Laboratory for Molecular Sciences, Key Laboratory of Bioorganic Chemistry and Molecular Engineering of Ministry of Education, Department of Chemical Biology, College of Chemistry and Molecular Engineering, Synthetic and Functional Biomolecules Center, and Peking-Tsinghua Center for Life Sciences, Peking University, Beijing, China. [5] National Laboratory of Biomacromolecules, CAS Center for Excellence in Biomacromolecules, Institute of Biophysics, Chinese Academy of Sciences, Beijing 100101, China. [6] Department of Microbiology and Biotechnology, College of Life Sciences, Northeast Agricultural University, No. 600 Changjiang Road, Xiangfang District, Harbin 150030, China. [7] Beijing StoneWise Technology Co Ltd., Haidian District, Beijing, China. [8] University of Chinese Academy of Sciences, Beijing 100049, China. [9] These authors contributed equally: Jiangtao Zhang, Yiqiang Shi, Junping Fan, Huiwen Chen. ✉email: jiankeg@hust.edu.cn; huangz@hsc.pku.edu.cn; jiangdh@iphy.ac.cn

on channels play a fundamental role in electrical signaling, which is crucial for numerous physiological processes including neuronal excitability, muscle contraction, secretion and perception of environmental changes[1–4]. Voltage-gated sodium ($Na_V$) and potassium channels work in concert to generate action potentials in electrical excitable cells[2]. These channels activate in response to depolarizing stimuli and inactivate rapidly to terminate ion flux. Both activation and inactivation are critical for tuning cellular excitability[1]. Dysfunction of either process causes abnormal channel function and leads to life-threatening diseases[5–7].

The four-domain $Na_V$ channel from eukaryotes usually inactivates within a few milliseconds[8]. Extensive studies had identified a three-residue hydrophobic motif, Ile-Phe-Met (IFM), located in the intracellular linker between domain III ($D_{III}$) and $D_{IV}$, which is responsible for the fast inactivation[4,8–10]. By contrast, prokaryotic $Na_V$ channels lack the IFM-motif and have slow inactivation over hundreds of milliseconds[11,12]. The fast N-type inactivation of potassium channels is controlled by their N-terminus or by a cytoplasmic auxiliary subunit, which are thought to work through a "ball-and-chain" mechanism[13–17]. Recent structural advances revealed that the IFM-motif of $Na_V$ channels serves as a hydrophobic latch that allosterically closes the activation gate[18–21]. The IFM-motif binds to a hydrophobic pocket adjacent to the activation gate, and its binding shifts the pore-lining S6 helices to close the gate. A recent structural study revealed that the N-terminus of the potassium channel MthK functions as a tethered "ball" physically blocks the open activation gate[13]. Those structural observations highlighted the distinct mechanisms for fast inactivation of eukaryotic metazoan $Na_V$ and potassium channels.

Surprisingly, despite lacking the signature IFM-motif, a family of eukaryotic protozoan homotetrameric $Na_V$ channels from the ubiquitous marine plants coccolithophores *Emiliania huxleyi* and *Scyphosphaera apsteinii*, exhibit fast inactivation property on the millisecond timescale similar to human $Na_V$ channels[22,23], suggesting an unknown alternative mechanism for $Na_V$ channel fast inactivation. Here, we employed cryo-electron microscopy (cryo-EM) and electrophysiological voltage clamp approaches to investigate the molecular mechanism for fast inactivation of the sodium channel $Na_V$Eh from the coccolithophore *Emiliania huxleyi*. Our results reveal an unexpected structural basis for the N-type fast inactivation of an $Na_V$ channel, mediated by its N-terminus and unrelated to the IFM-motif.

## Results

**Functional analysis of $Na_V$Eh and structure determination.** The sodium channel $Na_V$Eh from *Emiliania huxleyi* is composed of a N-terminal helix (N-helix), 6 transmembrane segments (S1-S6) and a C-terminal EF-hand like domain (EF-L) with a total of 542 amino acid residues (Fig. 1a), suggesting the channel is formed in a homotetrameric fashion. It shares amino acid sequence identity of 27% with the bacterial $Na_V$ channel $Na_V$Ab and 21% with human $Na_V$1.7, respectively (Supplementary Fig. 1), indicating $Na_V$Eh is more closely related to bacterial $Na_V$ channels. The gene encoding $Na_V$Eh was subcloned into a HEK293-F cell expression vector fused with a green fluorescent protein (GFP) at the C-terminus to facilitate tracing protein expression. We examined the functional characteristics of $Na_V$Eh expressed in HEK293T cell by whole-cell voltage clamp. $Na_V$Eh generated rapid inward currents in response to depolarizing pulses and became inactivated within 5 ms (Fig. 1b). The $V_{1/2}$ for voltage-dependent activation and steady-state fast inactivation are $-61.5 \pm 2.0$ mV ($n = 15$) and $-94.4 \pm 2.1$ mV ($n = 9$), respectively (Fig. 1b). $Na_V$Eh displayed fast inactivation that closely resembles the asymmetric four-domain eukaryotic $Na_V$ channels[20,24,25], and differs markedly

from the homotetrameric $Na_V$ channels from prokaryotes that lack fast inactivation[11,12]. Surprisingly, no IFM-motif like sequence was found in the $Na_V$Eh protein sequence (Supplementary Fig. 1).

To investigate the molecular mechanism underlying fast inactivation of $Na_V$Eh independent of the IFM-motif, we purified a homogeneous sample of $Na_V$Eh in detergents (Supplementary Fig. 2a), and performed cryo-EM single-particle analysis of the purified sample, yielding a final reconstruction map at an overall resolution of 2.8 Å (Fig. 1c and Supplementary Fig. 2b). The excellent density map allowed accurate de novo model building of the region W69-D358 (Fig. 1d and Supplementary Fig. 3). Four blobs of globular density located in the cytosol may belong to the C-terminal EF-L domain (Fig. 1c); however, the map quality of this part was not sufficient to build a reliable model for the EF-L domain. Strikingly, a strong tubular density of ~26 Å in length was observed half-embedded in the intracellular activation gate of $Na_V$Eh (Fig. 1c). The tube-like density consistently emerged when map reconstruction was performed with either C1- or C4-symmetry imposed (Fig. 1c and Supplementary Fig. 3c), indicating that the intrinsic density may belong to a helix that extends into the activation gate.

**Architecture of $Na_V$Eh and its open activation gate.** The $Na_V$Eh structure is assembled by four identical subunits in domain-swapped organization (Fig. 1d). Each subunit is composed of a voltage-sensing domain (VSD, S1-S4) and a pore module (PM, S5-S6). The transmembrane core region of $Na_V$Eh resembles the bacterial $Na_V$Ab[26] and human $Na_V$1.5[25] structures with root mean square deviation (RMSD) of 2.4 Å and 2.9 Å respectively, highlighting the conserved architecture of $Na_V$ channels across a wide range of species. Distinct from $Na_V$Ab, $Na_V$Eh has an extracellular loop (ECL) between S5 helix and pore-helix 1 (P1), extending the vestibule ~25 Å tall above the ion selectivity filter (SF) (Fig. 1a, d). Each ECL consists of a pair of anti-parallel beta sheets and two short helices projecting above the PM, which form extensive interactions with adjacent ECL to stabilize the vestibule (Fig. 2a, b). For example, carbonyl oxygen atoms of G237 and G280 form polar interactions with adjacent R254 and G271, respectively (Fig. 2b). A blob of strong density for a solvent molecule, mediates electrostatic interaction between E285 and N251 of neighboring subunit (Fig. 2b, red sphere). Notably, the ECL is in rich of acidic residues, which generate a strong electronegative surface and provide additional anionic coordination sites for cations (Fig. 2c). The four ECLs form a funnel-shaped vestibule with a diameter of 23.3 Å on the extracellular end, which narrows to 7.3 Å at the SF (Fig. 2d). Interestingly, such ECLs are not found in prokaryotic $Na_V$ channels (Fig. 3a), though significant differences between the ECLs of $Na_V$Eh and mammalian $Na_V$ channels (Fig. 3b, c) remain. At the bottom of the vestibule, a short loop of [303]TGESWSE[309] between P1 and P2 helices constitute the SF of $Na_V$Eh. The SF loop is highly conserved with the [175]TLESWSM[181] loop of its prokaryotic analog $Na_V$Ab[26]. Four Glu residues at the +3 position mainly determine sodium selectivity by forming a high-field strength site for dehydrating $Na^+$ ions[26] (Figs. 2c, d and 3d). However, the symmetric SF of $Na_V$Eh differs from the asymmetric SF of metazoan $Na_V$ channels, which feature the signature DEKA sequence[27]. Structure superposition revealed that the square SF of $Na_V$Eh is nearly identical to that of $Na_V$Ab with van der Waals diameter of 4.6 Å (Figs. 2e and 3d), indicating the SF of $Na_V$Eh is more closely related to $Na_V$Ab than to heterotetrameric eukaryotic Nav channels. Because the Lys from $D_{III}$ of the DEKA sequence of heterotetrameric Nav channels was consistently found pointing inside the SF[18–20,25,28], the asymmetric SF of human $Na_V$ channel is shorter than $Na_V$Eh in

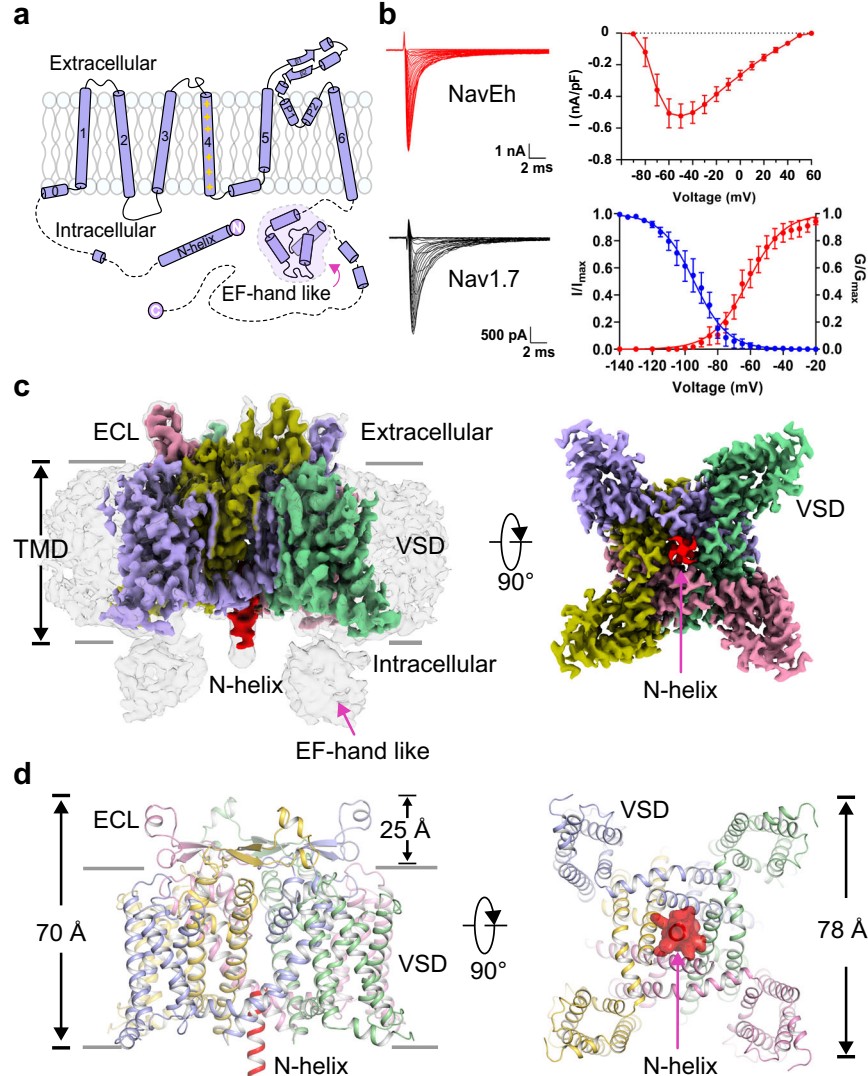

**Fig. 1 Cryo-EM structure of Na$_V$Eh. a** Topology of Na$_V$Eh. Red plus signs represent gating charges on S4 helix. The dashed lines indicate unsolved region. **b** Electrophysiological characterization of the Na$_V$Eh. A family of sodium currents conducted by Na$_V$Eh (red) and Na$_V$1.7 (black), respectively. Average peak current-voltage curve is shown (right top), where currents were normalized to cell capacitance. Normalized conductance-voltage (G/V) relationship (Red circles) and steady-state inactivation (Blue circles) of Na$_V$Eh. For measuring current-voltage curve, currents were measured from Na$_V$Eh transfected HEK293T cells with 100-ms depolarizing pulses between −100 mV and 60 mV in steps of 10 mV from a holding potential of −150 mV. For measuring G/V curve, currents were measured with 100-ms depolarizing pulses between −100 mV and 20 mV in steps of 5 mV from a holding potential of −150 mV. For measuring steady-state inactivation, prepulse potentials between −140 mV and −20 mV in 5 mV increments for 500-ms were applied and followed by a 50-ms test pulse at 0 mV. The Boltzmann fitted data yielded activation $V_{1/2}$ = −61.5 ± 2.1 mV ($n = 15$) and steady-state inactivation $V_{1/2}$ = −94.4 ± 2.1 mV ($n = 9$). Data are mean +/− SEM. **c** The cryo-EM density map of Na$_V$Eh. The four subunits and N-helix are colored in purple, yellow, pink, green and red, respectively. **d** Cartoon representation of Na$_V$Eh. The N-helix is shown in half-transparent surface for the right panel viewed from intracellular side. Source data are provided.

one dimension (Fig. 3e). Despite the conformational and compositional differences, all three SFs confer Na$^+$ selectivity[11,22,27]. Although possible models have been proposed to explain the sodium selectivity[20,26,29–32], the exact structural mechanisms for discriminating Na$^+$ need further investigation.

The VSD of Na$_V$Eh was resolved in an activated conformation, similar to other Na$_V$ channel structures[19,20,26]. The gating charge carrier S4 helix adopts 3$_{10}$ helical conformation, with three of its five gating charges in the activated "up" conformation above the hydrophobic-constriction site (HCS) (Fig. 3f). Two gating charges are stabilized by intracellular negatively charged clusters (INC) below the HCS, suggesting the VSD of Na$_V$Eh is less fully activated than that of Na$_V$Ab or VSD$_I$, VSD$_{II}$, and VSD$_{III}$ of human Na$_V$ channels (Fig. 4f–i). Activation of the VSDs generally

causes pore opening or channel transition to a non-conductive inactivated state. We next calculated the pore radius of Na$_V$Eh at the intracellular activation gate, excluding the N-helix. As illustrated in Fig. 2d, the van der Waals diameter for the activation gate of Na$_V$Eh is ~8 Å, wider than the size of hydrated Na$^+$ (7.2 Å)[33], indicating that the gate of Na$_V$Eh is fully open. A closer look at the gate revealed that the gate is constricted by four L350 residues, whose distance is 11.5 Å measured from the side-chain distal carbon atoms of opposing residues (Fig. 2f). In fact, structural superposition suggests the gate of Na$_V$Eh is slightly wider than the open gate of Na$_V$Ab[34] and rat Na$_V$1.5[21] (Fig. 2e, f). Collectively, the activated VSD, open activation gate and the presence of a helix blocking the open gate indicate the Na$_V$Eh structure was captured in its open-inactivated state.

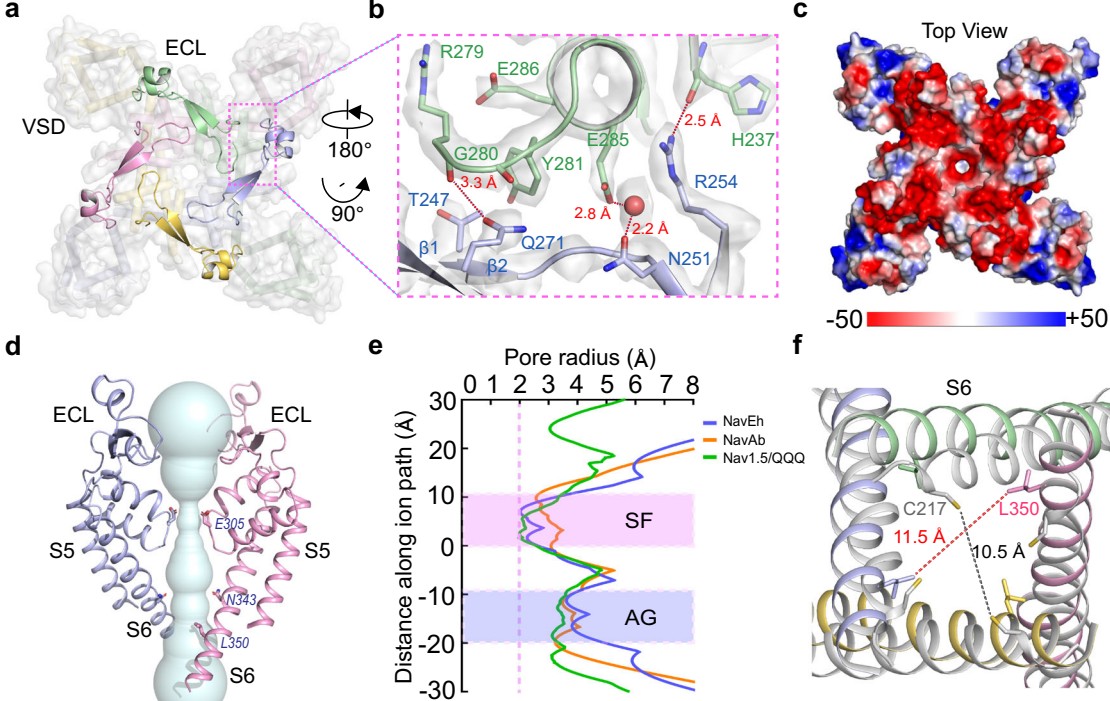

**Fig. 2 The unique ECL and open gate of NaᵥEh. a** The ECLs of NaᵥEh form external vestibule. Cartoon representation of the ECLs viewed from extracellular side. The transmembrane region shown in half-transparent surface. Pink dashed square indicates area shown in panel **b**. **b** A close-up view of interactions between the ECLs. The side-chains of key interacting residues are depicted in sticks. The red sphere represents a solvent molecule. The EM-density map is contoured at 5 σ. **c** Electrostatic potential of NaᵥEh colored from −50 to 50 kT (red to blue). **d** Ion path of NaᵥEh calculated by HOLE. Residues at the constriction sites of selectivity filter (SF) and activation gate (AG) shown side-chains in sticks. Only two opposing subunits shown for clarity. **e** Pore radii of NaᵥEh from panel **d**. As a comparison, pore radii of open NaᵥAb (PDB code: 5VB8, brown) and open Naᵥ1.5 (PDB code: 7FBS, green) are presented. **f** Gate superposition of NaᵥEh and open NaᵥAb (PDB code: 5VB8, colored in white).

**The N-helix mediates fast inactivation of NaᵥEh.** The helical density blocking the open gate coincides with the C4-symmetric axis of NaᵥEh; consequently, the density was averaged during the final refinement with C4-symmetry imposed (Fig. 1c and Supplementary Fig. 3c). To define which part of NaᵥEh actually forms the observed density, we performed electrophysiological studies of NaᵥEh variants with deletion or mutation. Deletion of the C-terminal P518-V542 (NaᵥEh$^{\Delta518-542}$) caused negligible effect on fast inactivation or activation compared to the wild-type (NaᵥEh$^{WT}$) (Supplementary Fig. 4a, b). By sharp contrast, deletion of N-terminal I2-R13 (NaᵥEh$^{\Delta2-13}$) completely abolished the fast inactivation (Fig. 4a and Supplementary Fig. 4b). In addition, we found that the fast inactivation of the NaᵥEh$^{\Delta2-13}$ can be partially restored when intracellularly applying a synthetic polypeptide of the N-helix (peptide$^{2-13}$) in a concentration-dependent manner (Supplementary Fig. 4b). At high concentration of 200 μM, the peptide$^{2-13}$ can restore the fast inactivation of ~88.6 ± 6.1% ($n = 9$) when test pulse was held at −35 mV (Fig. 4a). These results strongly indicate that the N-helix is responsible for the fast inactivation of NaᵥEh. To confirm that the N-helix blocks the gate, we purified a NaᵥEh$^{\Delta2-13}$ protein sample and solved its cryo-EM structure at 4.0 Å resolution (Supplementary Fig. 5). The EM map of NaᵥEh$^{\Delta2-13}$ clearly showed a hollow gate without any visible density (Supplementary Fig. 5c, d), which confirms that the N-helix indeed binds in the activation gate and blocks it.

Sequence analysis shows that the first two helical-turns of the N-helix are composed of hydrophobic or small side-chain residues followed by five consecutive positively charged residues of Arg9-Arg13 (Arg-cluster) (Fig. 4b). We found that the N-helix can be neatly fitted into the density (Supplementary Fig. 3c). In particular, the first two helical-turns are embedded inside the open activation

gate without clashing with the gate (Fig. 4c). Furthermore, we noticed that the outer mouth of the activation gate is rich in negatively charged residues (Fig. 4b, d); therefore, the Arg-cluster can form multiple electrostatic interactions with the negatively charged residues on the four S6 helices (Fig. 4c, d). We hypothesize that the Arg-cluster in the N-helix forms a pre-docking complex for fast inactivation by interacting loosely with the negatively charged outer mouth of the gate and moving the N-helix close to it. In this pre-docking position, the N-helix can plug the gate rapidly after gate opening. To validate this hypothesis, we mutated the Arg9-Arg13 to five Glu (NaᵥEh$^{nE5}$) and examined the fast inactivation property of the mutants. Strikingly, the fast inactivation of NaᵥEh$^{nE5}$ was completely removed by the mutations (Fig. 4a). These results clearly demonstrate that the electrostatic interactions play critical role in the fast inactivation process.

The N-terminus mediated inactivation in potassium channels is well-studied and often termed the "ball-and-chain" mechanism[13,15,17,35–37]. The fast inactivation of potassium channels is removed after deletion of the N-terminus[15,17,37] and restored by intracellular application of the N-terminal polypeptide[17,37,38], similar to the results we found for the Naᵥ channel NaᵥEh (Fig. 4a). Sequence alignment reveals that the N-terminus of the three coccolithophore Naᵥ channels, the potassium channels and their accessory subunits feature a conserved sequence motif with a short region of hydrophobic residues followed by a positively charged cluster poised to enter and block the activation gate (Supplementary Fig. 6a, b). Recent progress from the cryo-EM structure of a calcium-gated prokaryotic potassium channel MthK revealed the structural basis for its N-type inactivation[13]. Even though the density map was resolved at medium resolution that did not allow side-chain assignment, it clearly showed an N-terminal

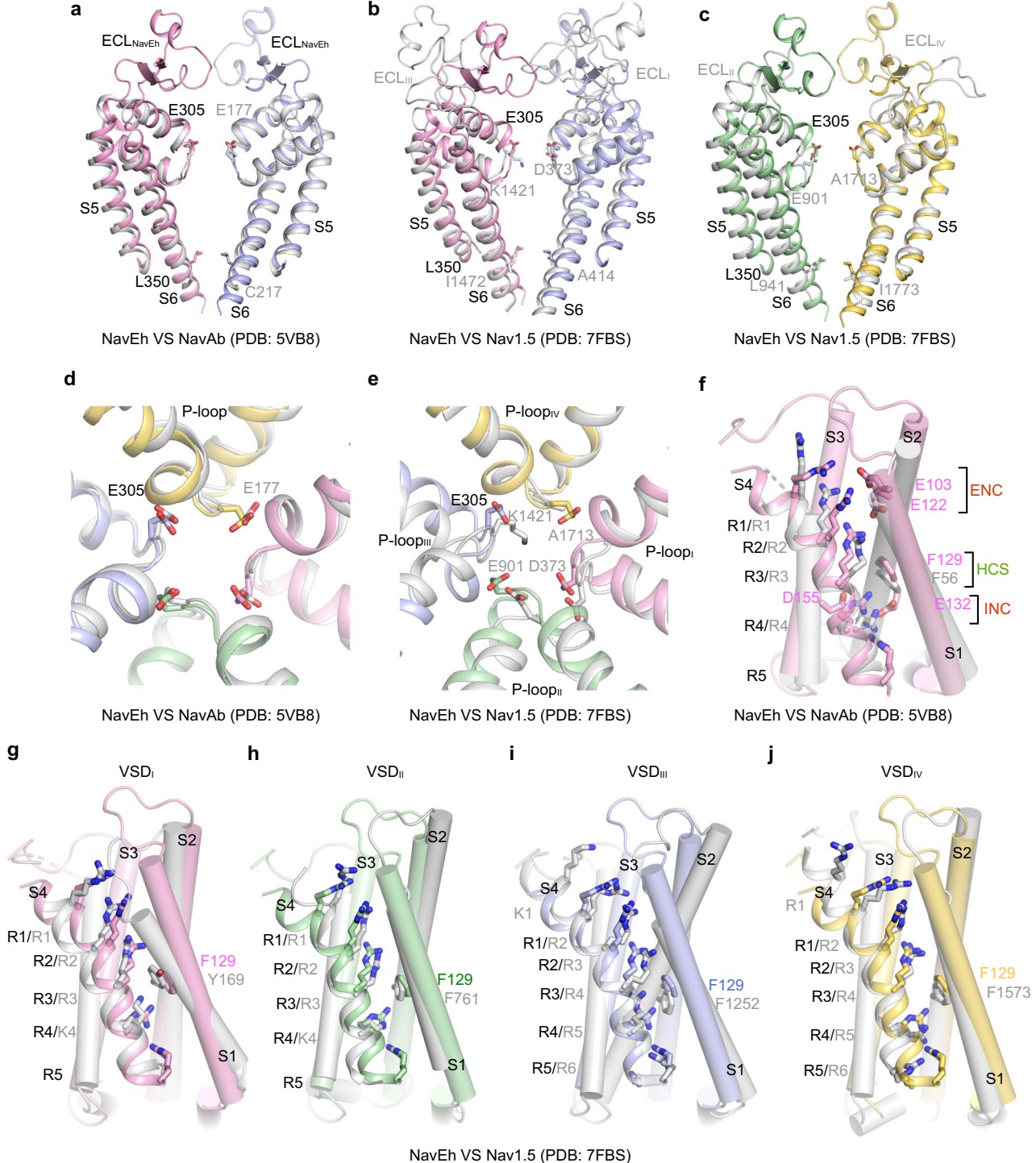

**Fig. 3 Structural comparison of Na$_V$Eh with Na$_V$Ab and Na$_V$1.5. a** Pore domain comparison between Na$_V$Eh (pink and light blue) and Na$_V$Ab (PDB: 5VB8, gray), S5 and S6 helices were used for superimposition. **b, c** Pore domain comparison between Na$_V$Eh and Na$_V$1.5 (PDB: 7FBS, gray). **d, e** Selectivity filter comparison of Na$_V$Eh with Na$_V$Ab and Na$_V$1.5, respectively. **f** VSD comparison of Na$_V$Eh and Na$_V$Ab (gray). The extracellular negatively charged clusters (ENC) and intracellular negatively charged clusters (INC), hydrophobic-constriction site (HCS) were colored in red and green, respectively. **g–j** VSD comparison between Na$_V$Eh and domain I–IV of Na$_V$1.5 (gray), respectively.

helical-like density inserted into the open activation gate. Superposition of the open gates of our Na$_V$Eh and the MthK (PDB code 6U68)[13] shows that both gates have a helix inserted in the middle; however, the gate of MthK is ~1.5 Å wider than that of Na$_V$Eh (Supplementary Fig. 6c, d). The smaller gate of Na$_V$Eh is caused by the tight corral formed by four S4-S5 linker helices, which is absent in the MthK channel (Fig. 1d and Supplementary Fig. 6c).

Nevertheless, these observations indicate that the N-type fast inactivation of Na$_V$Eh is similar to the N-type inactivation of potassium channels in mechanism.

**Unexpected N-type fast inactivation of Na$_V$ channels.** Fast inactivation is the hallmark feature of eukaryotic Na$_V$ channels. Structures of eukaryotic Na$_V$ channels have established structural

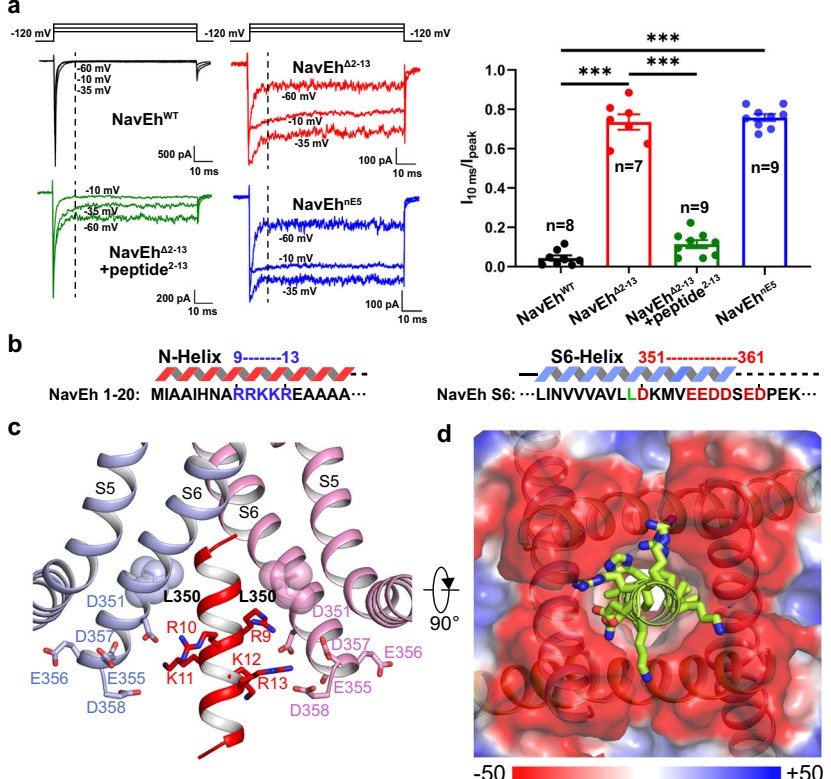

**Fig. 4 N-helix determines fast inactivation of Na$_V$Eh. a** Functional characteristics of Na$_V$Eh variants. Representative current traces measured at −65 mV, −35 mV, and −10 mV for Na$_V$Eh$^{WT}$, Na$_V$Eh$^{\Delta2-13}$, Na$_V$Eh$^{\Delta2-13}$+peptide$^{2-13}$ and Na$_V$Eh$^{nE5}$, respectively. The ratio of current at 10 ms (I$_{10\ ms}$) over peak current (I$_{peak}$) measured at −35 mV were shown in the right panel. Each dot represents a single-cell recording. Significances were determined using two-sided, unpaired $t$-test. ***$P$ = 1.9 × 10$^{-10}$ (Na$_V$Eh$^{\Delta2-13}$, $n$ = 7); ***$P$ = 4.0 × 10$^{-15}$ (Na$_V$Eh$^{\Delta2-13}$ + peptide$^{2-13}$, $n$ = 9); ***$P$ = 4.9 × 10$^{-10}$ (Na$_V$Eh$^{nE5}$, $n$ = 9). Data are mean +/− SEM. **b** Sequence of Na$_V$Eh N-helix and S6 helix. The positively charged cluster on N-helix and negatively charged cluster on S6 shown in blue and red, respectively. **c** A close-up view of the N-helix sticks in the gate. Key residues shown side-chains in sticks. Only two opposing subunits are shown for clarity. **d** Electrostatic potential of the Na$_V$Eh gate from −50 to 50 kT (red to blue). The yellow N-helix shown side-chains in sticks. Source data are provided.

basis for understanding the molecular mechanism of fast inactivation mediated by the IFM-motif[18–20,25,28]. The IFM-motif folds into the channel structure and binds tightly to a hydrophobic receptor site adjacent to the S6$_{IV}$ helix, which shifts the S6$_{IV}$ helix and allosterically closes the activation gate (Fig. 5a). Release of the IFM-motif from its receptor site leads to pore opening[21]. This unique local allosteric inactivation mechanism for Na$_V$ channels not only can rapidly terminate sodium influx to prevent overactivity (Fig. 5c, $\tau_{inact}$ = 2.8 ± 1.4 msec at −10 mV), but can also ensure that the channels can quickly recover from fast inactivation in order to permit repetitive firing in nerve and muscle cells[39,40] (Fig. 5e, $\tau_{fast}$ = 10.1 ± 2.3 msec). Unlike the canonical IFM-motif mediated inactivation, our Na$_V$Eh structure demonstrates an alternative mechanism for fast inactivation of Na$_V$ channels that is fundamentally different. The N-helix of Na$_V$Eh plugs into its open activation gate and physically blocks it (Fig. 5b), similar to the "ball-and-chain" mechanism observed in potassium channels[13]. Interestingly, the fast inactivation time course of Na$_V$Eh is comparable to the mammalian sodium channels (Fig. 5d, $\tau_{inact}$ = 1.3 ± 0.1 msec at −10 mV), indicating that the IFM-motif is not a prerequisite for the fast kinetics of Na$_V$ channel inactivation. Because IFM-motif mediated fast inactivation depends on activation of VSD$_{IV}$[19,20,41], its time course is voltage-dependent (Fig. 5c). In contrast, the time course of fast inactivation for Na$_V$Eh is independent of voltage (Fig. 5d), suggesting that the N-helix mediated fast inactivation is open-state inactivation. Strikingly, the recovery of Na$_V$Eh from fast inactivation is about 157-fold slower than human Na$_V$1.7 (Fig. 5f,

$\tau_{fast}$ = 1584 ± 473 msec). The recovery rate is even slower than human Na$_V$1.7 and Na$_V$1.8 recovery from slow inactivation[42], which were reported to be less than 1 s. Our Na$_V$Eh structure provides key structural information that explains the dramatic differences in recovery rate between Na$_V$Eh and Na$_V$1.7 (Fig. 5a, b). The binding of the IFM-motif to its receptor site buries a total of 866-Å$^2$ solvent accessible surface. However, the N-helix embedded inside the activation gate buries a total surface of 1688-Å$^2$, almost 2-fold greater than the IFM-motif. In addition, the multiple electrostatic interactions between the Arg-cluster on the N-helix and the negative charges on the S6 helices further strengthen the binding of the N-helix (Fig. 4c, d). The stronger binding interactions of the N-helix indicate that the energy barrier for releasing the N-helix from the open gate would be much higher than releasing the IFM-motif from its receptor site.

## Discussion

In this study, we presented high-resolution cryo-EM structure of the eukaryotic sodium channel Na$_V$Eh from the unicellular phytoplankton *Emiliania huxleyi*. The Na$_V$Eh structure shares a conserved core region with Na$_V$ channels from bacteria and mammals[19,20,25,26,28], but it is more closely related to the bacterial Na$_V$ channels, especially in its homotetrameric assembly and selectivity filter. However, Na$_V$Eh possesses an additional ECL domain and intracellular EF-L domain that might regulate channel function compared to Na$_V$Ab. More importantly, Na$_V$

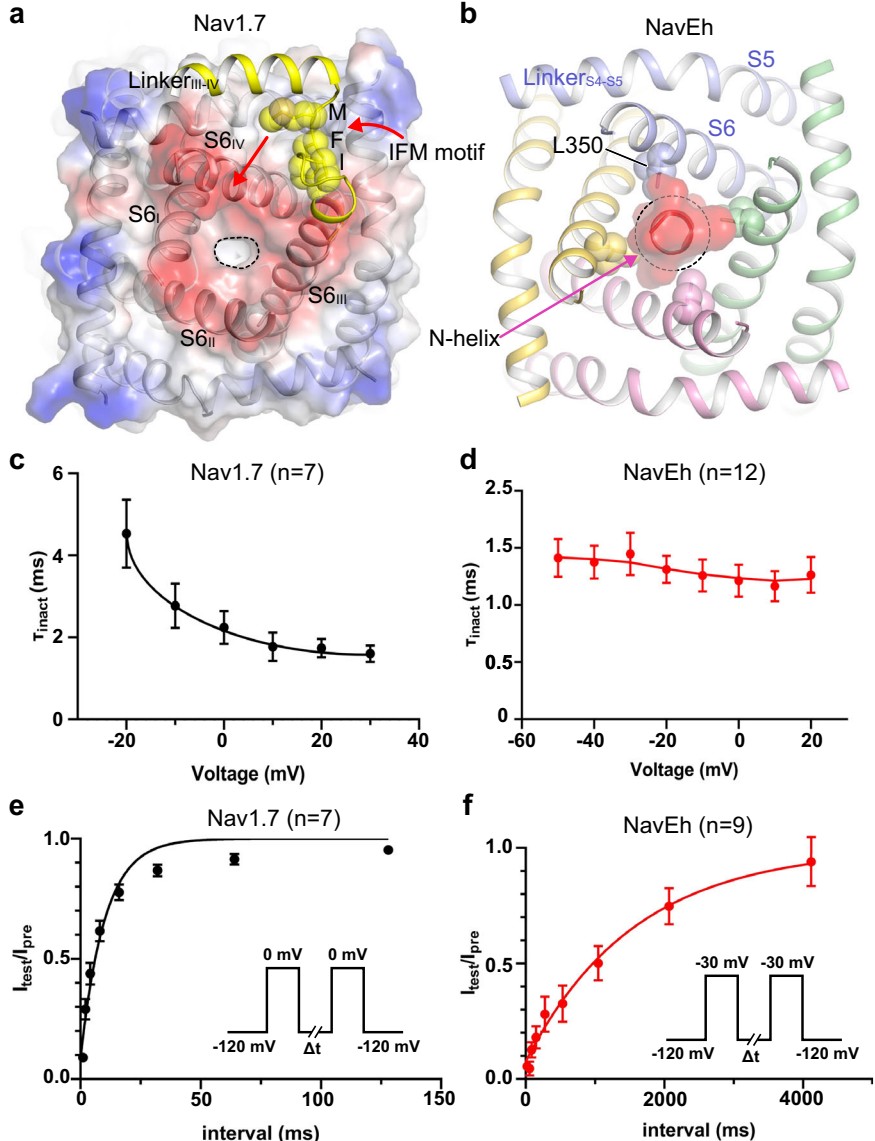

**Fig. 5 Comparison of N-helix and IFM-motif mediated inactivation. a**, **b** Cartoon representation of the IFM-motif mediated allosteric inactivation of human $Na_V1.7$ (PDB: 6j8j) and the N-helix mediated inactivation of $Na_VEh$. IFM-motif shown in spheres and colored in yellow. Dashed black circle indicates the gate size. **c**, **d** Fast inactivation time course of $Na_V1.7$ ($n = 7$) and $Na_VEh$ ($n = 12$). The time constant was plotted to the test voltages. **e**, **f** The time course for recovery from fast inactivation of $Na_V1.7$ ($n = 7$) and $Na_VEh$ ($n = 9$). For each point, data are means $+/-$SEM. Source data are provided.

channels from the unicellular phytoplankton like $Na_VEh$ have gained the ability of fast inactivation similar to metazoan $Na_V$ channels, unlike its prokaryotic homologs that are inactivated at much slower pace, possibly through a C-type slow inactivation mechanism[43–46]. We identified the N-helix of $Na_VEh$ as the key determinant for its fast inactivation, revealing the N-type like fast inactivation for $Na_V$ channels that is similar to the "ball-and-chain" mechanism for potassium channels[13,15,17]. Our high-resolution structure, complemented with electrophysiological results, provides detailed mechanistic insights into N-helix mediated inactivation, and structural information that potentially explains its similar fast inactivation but much slower recovery from fast inactivation compared with the canonical IFM-motif mediated fast inactivation. The fascinating marine plants coccolithophores are critical for the marine ecology and are highly relevant to climate change[47,48]. The fast inactivation of $Na_VEh$ may be important for the unicellular phytoplankton to tolerate the high concentration of sodium in the living

environment[22], but the slow recovery may prevent use of this mechanism in metazoan that require high-frequency electrical signaling. It will be of great interest to determine the precise physiological role of $Na_VEh$ and its unique fast-inactivation process in coccolithophores and other single-celled organisms.

## Methods

**Whole-cell voltage-clamp recordings of $Na_VEh$ in HEK293 T cells**.
HEK293 T cells were cultured with Dulbecco's Modified Eagle Medium (DMEM) (Gibco, USA) supplemented with 10% (v/v) fetal bovine serum (FBS, PAN-Biotech, Germany) at 37 °C with 5% $CO_2$. HEK293 T cells were transfected with plasmids of $Na_VEh$ WT or mutants using Lipofectamine 2000 Reagent (Thermo Fisher Scientific, USA) for 12 h. Experiments were performed 12–24 h post transfection at room temperature (22–25 °C). In brief, cells were placed on a glass chamber containing 140 mM NaCl, 3 mM KCl, 10 mM HEPES, 10 mM D-Glucose, 1 mM $MgCl_2$, 1 mM $CaCl_2$, (pH = 7.3 with NaOH and osmolarity of ~310 mOsm/L). Whole-cell voltage-clamp recordings were made from isolated, GFP-positive cells using 1.5 ~ 2.5 MΩ fire polished pipettes (Sutter Instrument, USA) filled with standard internal solution, containing 140 mM CsF, 10 mM HEPES, 1 mM EGTA, 10 mM NaCl, (pH = 7.3 with CsOH and osmolarity of ~300 mOsm/L). Whole-cell

currents were recorded using an EPC-10 amplifier (HEKA Elektronik, Germany) at 20 kHz sample rate and was low pass filtered at 10 kHz. The series resistance was 2~6 MΩ and was compensated 70~90%. The data was acquired by PatchMaster program (HEKA Elektronik, Germany).

To characterize the activation properties of $Na_V Eh$ channels, cells were held at −150 mV and then a series of 100 ms voltage steps from −100 mV to +20 mV (5 mV increments) were applied. The time constant (τ) of fast inactivation was from single exponential fits of $Na_V Eh$ activation in response to depolarization using this voltage protocol. The fast inactivation properties of $Na_V Eh$ channels were assessed with a 500 ms holding-voltages ranging from −140 mV to −20 mV (5 mV increments) followed by a 50 ms test pulse at −50 mV. The recovery properties were assessed by a double-pulse protocol using a varying interval between the two voltage pulses. Holding potential was −150 mV and prepulse was −50 mV for 20 ms, followed by a recovery test pulse of −50 mV for 5 ms at 32–4096 ms. The currents elicited by the test pulse were normalized to construct the recovery curve.

As for the voltage-clamp recording analyses, all data were reported as mean ± SEM. Data analyses were performed using Origin 2019b (OriginLab, USA), Excel 2016 (Microsoft, USA), and GraphPad Prism 8.0.2 (GraphPad Software, USA).

Steady-state activation curves were generated using a Boltzmann equation.

$$\frac{G}{G_{max}} = \frac{1}{1 + \exp[(V - V_{0.5})/k]} \tag{1}$$

where $G$ is the conductance, $G_{max}$ is the maximal conductance of $Na_V Eh$ during the protocol, $V$ is the test potential, $V_{0.5}$ is the half-maximal activation potential and $k$ is the slope

Fast inactivation curves were generated using a Boltzmann equation.

$$\frac{I}{I_{max}} = \frac{1}{1 + \exp[(V - V_{0.5})/k]} \tag{2}$$

where $I$ is the current at indicated test pulse, $I_{max}$ is the maximal current of $Na_V Eh$ activation during test-pulse, $V$ is the test potential, $V_{0.5}$ is the half-maximal inactivation potential and $k$ is the slope factor.

Recovery curves from fast inactivation were fit using a single exponential of the following equation.

$$\frac{I_{test}}{I_{pre}} = (y_0 - 1) * \exp\left(-\frac{t}{\tau}\right) + 1 \tag{3}$$

where $I_{pre}$ is the current at prepulse, $I_{test}$ is the current at test pulse, y0 is the non-inactivated current at the first pulse, t is the delay time between prepulse and test-pulse, and τ is the time constant of recovery from fast inactivation.

**Expression and purification of $Na_V Eh$.** The codon-optimized gene encoding NavEh (*Emiliania huxleyi*) was synthesized and was subcloned into the modified pEG BacMam vector[49] (Supplementary Table 1). In order to monitor protein expression and purification, a green fluorescent protein (GFP) and a Twin-Strep tag were fused to the C-terminus of NavEh. All constructs were confirmed by DNA sequencing. HEK293-F cells were cultured with Freestyle 293 medium at 37 °C, supplied with 5% (v/v) $CO_2$. When the cell density reached $2.5 \times 10^6$ cells/mL, a mixture (3:1) of expression plasmid and polyethylenimine (Polysciences) was added to the cell culture following a standard transfection protocol. After 12 h, sodium butyrate (Sigma, USA) was added to the culture at a final concentration of 10 mM, and the cells were incubated for another 48 h before harvesting.

For purification, the cell pellets were resuspended in Buffer A containing 20 mM HEPES pH 7.5, 150 mM NaCl, 2 mM β-mercaptoethanol (β-ME), and a protease inhibitor cocktail including 1 mM phenylmethylsulfonic acid Acyl fluoride (PMSF), 0.8 μM pepstatin, 2 μM leupeptin, 2 μM aprotinin, and 1 mM benzamidine. Then cells were disrupted with a Dounce homogenizer and membrane fractions were enriched by ultracentrifugation at 36,900 rpm for 40 min. Subsequently, the membrane protein fraction was resuspended in buffer B (buffer A supplemented with 1% (w/v) n-dodecyl-β-D-maltoside (DDM), 0.15% (w/v) cholesterol Hemisuccinate (CHS), 5 mM $MgCl_2$, and 1 mM ATP), and agitated at 4 °C for 2 h. The insoluble membrane fraction was removed by ultracentrifugation at 36,900 rpm for 40 min. Then the supernatant was incubated with Strep-Tactin beads (Smart-Lifesciences), which was pre-equilibrated with buffer C (buffer A supplemented with 5 mM $MgCl_2$, 5 mM ATP, and 0.06% (w/v) Glyco-diosgenin (GDN) (Anatrace)). Subsequently, the Streptactin beads were washed with 10 column volumes of buffer C and buffer D (buffer C without 5 mM $MgCl_2$ and 5 mM ATP), respectively. The protein was eluted by 5 ml buffer E (buffer D plus 5 mM desthiobiotin). The elution was concentrated and loaded onto Superose Increase 10/300 GL (GE Healthcare, USA) pre-equilibrated with 20 mM HEPES, 150 mM NaCl, 0.007% GDN (w/v), and 2 mM β-mercaptoethanol (β-ME), pH 7.5. Peak fractions were collected and concentrated to 7.8 mg/mL.

**Cryo-EM sample preparation and data collection.** Aliquots of 2.5 μL purified sample was placed on glow-discharged holey copper grids (Quantifoil Cu R1.2/1.3, 300 mesh), which were blotted for 2.5–3.5 s and plunge-frozen in liquid ethane cooled by liquid nitrogen using a FEI Mark IV Vitrobot at 4 °C with 100% humidity. All data were acquired using a Titan Krios transmission electron microscope operated at 300 kV, a Gatan K2 Summit direct detector and Gatan

Quantum GIF energy filter with a slit width of 20 eV. All movie stacks were automatically collected using SerialEM at a calibrated magnification of 105,000× with a physical pixel size of 1.04 Å (super-resolution mode). The defocus values were set from −1.2 to −2.2 μm. The dose rate was adjusted to 10 counts/pixel/s. A total of 1014 and 1119 movie stacks were collected for $Na_V Eh^{WT}$ and $Na_V Eh^{Δ2-13}$, respectively. Each movie stack was exposed for 6.4 s fractionated into 32 frames with a total dose of 60 e−/Å$^2$.

**Data processing.** All the movie stacks were motion-corrected, binned by 2-fold and dose-weighted using MotionCorr2[50], yielding a pixel size of 1.04 Å. Defocus values of each summed micrographs were estimated with Gctf[51]. A total of 299,062 and 579,023 particles were auto-picked for $Na_V Eh^{WT}$ and $Na_V Eh^{Δ2-13}$, respectively. All 2D classification, 3D classification, polishing, and CTF refinement were carried out in RELION3.0[52]. The detailed data processing flow was shown in Supplementary Figs. 2 and 5. The best class containing 61,065 and 64,407 particles for $Na_V Eh^{WT}$ and $Na_V Eh^{Δ2-13}$ were refined using cryoSPARC[53] to 2.83 Å and 4.02 Å resolution, respectively.

**Model building.** The predicted AlphaFold model of $Na_V Eh$ was fitted into the cryo-EM density map of $Na_V Eh$ using Chimera[54], manually checked, and corrected in COOT[55]. Then the resulting model were refined in Phenix[56]. The model vs. map FSC curve was calculated by Phenix.mtrage. The statistics of cryo-EM data collection and model refinement were summarized in Supplementary Table 2.

All figures were prepared with PyMOL (Schrödinger, LLC), and Prism 8.0.1 (GraphPad Software) and ChimeraX[57].

**Reporting summary.** Further information on research design is available in the Nature Research Reporting Summary linked to this article.

## Data availability

The data that support the findings of this study are available from the corresponding author upon reasonable request. The amino acid and gene sequences of NavEh (MMETSP transcriptomic datasets [https://www.bco-dmo.org/dataset/665427] ID: CAMPEP_0187654740, MMETSP0994-7) are provided in Supplementary Table 1. Atomic coordinates have been deposited in the Protein Data Bank under the accession code 7X5V (Na$_V$Eh), and the corresponding EM map has been deposited in the Electron Microscopy Data Bank under the accession number EMD-33016 (Na$_V$Eh). PDB accession codes used in this study are 5VB8 (NavAb), 7FBS (Nav1.5), 6J8J (Na$_V$1.7), and 6U68 (MthK). Source data of Figs. 1b, 4a, 5c–f, and Supplementary Figs. 2a and 4a are provided with this paper.

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

## Acknowledgements

We thank X. Huang, B. Zhu, X. Li, L. Chen, and other staff members at the Center for Biological Imaging (CBI), Core Facilities for Protein Science at the Institute of Biophysics, Chinese Academy of Science (IBP, CAS) for the support in cryo-EM data collection. We thank Prof. William A. Catterall, Prof. Ning Zheng, Prof. Yan Zhao, and Prof. Xuejun Cai Zhang for their helpful discussions, Bei Yang, Yan Wu, and Wei Fan for their research assistant service. This work is funded by Institute of Physics, Chinese Academy of Sciences (E0VK101 to D.J.), the National Natural Science Foundation of China (31871083 and 81371432 to Z.H., 32070031 to J.J., 82071851 to J.G.), the Chinese National Programs for Brain Science and Brain-like intelligence technology (2021ZD0202102 to Z.H.), and the program for HUST Academic Frontier Youth Team (5001170068 to J.G.).

## Author contributions

D.J. conceived and designed the experiments. J.Z. and Z.X. prepared sample for cryo-EM study and made all the constructs. J.Z. and J.F. collected cryo-EM data. H.C., J.Z. and B.H. processed the data, built and refined the models. J.Z., H.C., and Y.S. prepared figures. Y.S. collected the electrophysiology data. J.J., J.G., Z.H., J. F., and D.J. analyzed and interpreted the results. J.F. and D.J. wrote the paper, and all authors reviewed and revised the paper.

## Competing interests

The authors declare no competing interests.
