## [Peer Review File · Nature Communications]

N-type fast inactivation of a eukaryotic voltage-gated sodium channelReviewers' Comments:

Reviewer #1:

Remarks to the Author:

This very interesting manuscript presents the first cryo-EM structure of NavEh, from marine plants, coccolithophores, *Emiliana huxleyi*. These channels have been shown previously to be single domain, fast sodium-selective voltage-gated channels with sensitivity to calcium (Helliwell et al). Remarkably, they exhibit rapid activation and inactivation despite the absence of the IFM inactivation gate motif found in mammalian sodium channels. Here, the authors present cryo-EM results for purified homotetrameric channels. The data suggest that an N-terminal helix facilitates fast sodium current inactivation by physically blocking the activation gate, where it becomes stabilized by electrostatic interactions. Deletion of the N terminal helix, or mutation of key arginine residues proposed to form the docking electrostatic interactions, abolished fast sodium current inactivation and shifted the activation curve to the right. Taken together, these results reveal a novel, N-type structural basis for inactivation of a sodium channel that is similar to what has been shown previously for potassium channels. While the cryo-EM results are impressive, I found some issues with the functional characterization of the channel. Even though this functional work is in some ways a repeat of the Helliwell et al studies, the experiments are incomplete and additional data should be shown.

1. Voltage clamp protocols: The inclusion of F in the internal pipette sodium is advantageous in forming seals, however, it can create problems in terms of the stability of voltage-dependence, especially for inactivation, which shifts negative in a time dependent manner in the presence of F. Under these conditions, it is necessary to wait 10-15 minutes after forming the seal for the voltage-dependence of inactivation to stabilize. How long did the investigators wait for this to occur and how did they determine the stability of these data? This is a critical point since the focus of the study is current inactivation.
2. Filtering the data at 5 kHz can result in artificially slow sodium current. It would be better, and more in line with other published data, to filter at 8-10 kHz.
3. How do the authors know that the GFP epitope tag did not affect channel function?
4. Complete IV curves as well as current family traces should be provided in Fig. 1 to demonstrate proper voltage control. It may be necessary to lower the extracellular driving force to achieve control.
5. The authors should present a complete set of ion selectivity data for their NavEH channel expression. What is the reversal potential? Is there calcium dependence? What happens when sodium is replaced with NMDG? Is the current sensitive to TTX and STX?
6. The voltage dependence of inactivation curve shown in Fig. 1b appears to have at least two components and should be re-analyzed as such. Does this second component indicate calcium dependence of the current?
7. The activation curve (Fig. 1b) starts at -100 mV. The data acquired from -140 to -100 mV should be analyzed and included.
8. The $V_{1/2}$ values for activation and inactivation of NavEH are quite negative compared to mammalian sodium channels as well as the channels described by Helliwell et al (Current Biology). How do the authors explain this discrepancy?
9. How many independent transfection experiments are represented in Fig. 1b?
10. Full families of current traces and IV curves should be presented in Fig. 3a, instead of representative traces at -35 mV. In addition, the n numbers for this experiment are very low (n=4-6). Finally, how many independent transfection experiments are represented by these data? This comment also applies to the data in Fig. 4.
11. How many independent transfection experiments are represented by the data shown in Extended Fig. 5?
12. The authors should discuss potential mechanisms for the rightward shifts in activation shown in Extended Fig. 5.
13. Fast inactivation is only partially restored by the 2-13 peptide (Extended Fig. 5), rather than "almost fully restored," as stated in the text (line 172). The authors should discuss this result and at least provide a potential explanation.

14. Are NavEH currents modulated by mammalian sodium channel beta subunits?

Reviewer #2:

Remarks to the Author:

A manuscript by Jiangtao Zhang et al. describes the cryo-EM structure of NaVEh from the coccolithophore *Emiliana huxleyi*. NaVEh structure reveals a novel molecular gating mechanism of Nav channel fast inactivation that is similar to the N-terminus mediated inactivation in Kv channels. Experimental testing confirmed key observations about the molecular mechanism of NaVEh channel inactivation from the NaVEh structure. This study provides valuable structural insights into NaVEh channel inactivation compared to classical fast inactivation of eukaryotic Nav channels. The manuscript is very well written, and the data are presented clearly. I have only one minor comment:

1. Move Extended Data Fig. 4 as a new Figure in the main text – Figure 5.

Reviewer #3:

Remarks to the Author:

Voltage-gated ion channels play fundamental roles in many biological processes. They are present in all kingdoms of life, from bacteria to human, and have diverse inactivation mechanism. In this manuscript, Zhang et al discovered that NaVEh has an IFM-independent fast inactivation mechanism. High resolution structure of NaVEh and its structure without N-helix reveal that the N-helix plugs into the open pore and blocks the currents. Electrophysiological experiments confirmed the functional role of N-helix in fast inactivation. The work is well carried out and the manuscript is well-written. I have no major concerns but a few minor suggestions for the authors.

1. The authors might use tools such as BLAST to identify if similar N-terminal sequences exist in other ion channels. This might broaden the scope of the current research.
2. NaVEh is not a well-characterized ion channel. The authors claimed that there is EF hand domain in C terminus and the 2nd 3D class seems to have stronger densities for such helices. Would the resolution of the 2nd 3D class improved to a level where EF hand domain could be modeled? More importantly, are the channel behaviors modulated by calcium?
3. The binding of small N-helix to the 4-fold axis complicated the computation and map interpretation. Refinement with C1 symmetry would be mainly driven by the signal of channel. Therefore, is it possible to use symmetry expansion and focused classification with high "Tau numbers" but without alignment to separate the four conformations of N-helix?

Response to Reviewers' Comments

Reviewer #1 (Remarks to the Author):

This very interesting manuscript presents the first cryo-EM structure of NavEh, from marine plants, coccolithophores, *Emiliana huxleyi*. These channels have been shown previously to be single domain, fast sodium-selective voltage-gated channels with sensitivity to calcium (Helliwell et al). Remarkably, they exhibit rapid activation and inactivation despite the absence of the IFM inactivation gate motif found in mammalian sodium channels. Here, the authors present cryo-EM results for purified homotetrameric channels. The data suggest that an N-terminal helix facilitates fast sodium current inactivation by physically blocking the activation gate, where it becomes stabilized by electrostatic interactions. Deletion of the N terminal helix, or mutation of key arginine residues proposed to form the docking electrostatic interactions, abolished fast sodium current inactivation and shifted the activation curve to the right. Taken together, these results reveal a novel, N-type structural basis for inactivation of a sodium channel that is similar to what has been shown previously for potassium channels. While the cryo-EM results are impressive, I found some issues with the functional characterization of the channel. Even though this functional work is in some ways a repeat of the Helliwell et al studies, the experiments are incomplete and additional data should be shown.

Reply: We appreciate Reviewer 1's positive comments on the novelty and cryo-EM results of this study, and his/her suggestions for improving our manuscript. We are interested in Helliwell et al studies that reported a novel Na_v channel with fast inactivation but without the signature IFM-motif, so we decided to figure out what is the molecular mechanism for this sodium channel fast inactivation unrelated to the IFM-motif. We agree with Reviewer 1 that Helliwell et al studies are pretty thorough in terms of functional characterization of the channel, and it's unnecessary to repeat all the work. Thus, we highlight the cryo-EM structure revealing the first N-type fast inactivation for Na_v channels in this work. In addition, we identified that it is the N-terminal helix that mediates the fast inactivation of Na_vEh , which was not reported in previous studies. We have revised our manuscript and the functional experiments following Reviewer 1's suggestions, as described below.

1. Voltage clamp protocols: The inclusion of F in the internal pipette sodium is advantageous in forming seals, however, it can create problems in terms of the stability of voltage-dependence, especially for inactivation, which shifts negative in a time dependent manner in the presence of F. Under these conditions, it is necessary to wait 10-15 minutes after forming the seal for the voltage-dependence of inactivation to stabilize. How long did the investigators wait for this to occur and how did they determine the stability of these data? This is a critical point since the focus of the study is current inactivation.

Reply: We thank Reviewer 1's suggestion. We typically waited for 5-10 min after forming the seal before measuring currents. We have also re-collected data when waiting time at 5 min, 10 min and 15 min after forming the seal to test the stability of these data. As shown in **Figure A** below, the data are stable between the corresponding waiting times.

2. Filtering the data at 5 kHz can result in artificially slow sodium current. It would be better, and more in line with other published data, to filter at 8-10 kHz.

Reply: We thank Reviewer 1's comments. We examined our protocol for voltage-recordings and found that we have made a mistake when we copied the method section from previous paper. Actually, all the data were filtered at 10 kHz and we have corrected this in the revision.

3. How do the authors know that the GFP epitope tag did not affect channel function?

Reply: We thank Reviewer 1 for raising this point. A GFP-tag was useful to monitor the expression of NavEh in the plasmid transfected HEK293 cells. In order to test the effect of GFP-tag, we made a new construct of WT NavEh into a vector with an internal ribosome entry site (IRES) followed by a mCherry fluorescent protein (NavEh-IRES-mCherry). In this case, the transfected HEK293 cells express separate proteins of NavEh and mCherry fluorescent protein. We measured sodium currents from the NavEh-IRES-mCherry transfected HEK293 cells, which exhibit indistinguishable voltage-dependence of activation or inactivation to NavEh-GFP (See **Figure** below).

Comparison of the voltage-dependence of activation and fast-inactivation between NavEh-GFP and NavEh-IRES-mCherry. The currents were recorded from the NavEh-IRES-mCherry plasmid transfected HEK293 cells using the same protocols described in **Fig. 1b**. Data were acquired from four independent transfection experiments.

4. Complete IV curves as well as current family traces should be provided in Fig. 1 to demonstrate proper voltage control. It may be necessary to lower the extracellular driving force to achieve control.

Reply: Corrected as requested. The current-voltage (I-V) curves were added in **new Fig. 1b**.

New Figure 1b, Electrophysiological characterization of the NavEh. A family of sodium currents conducted by NavEh (red) and Nav1.7 (black), respectively. Average peak current-voltage curve is shown (right top), where currents were normalized to cell capacitance. Normalized conductance-voltage (G/V) relationship (Red circles) and steady-state inactivation (Blue circles) of NavEh. For measuring current-voltage curve, currents were measured from NavEh transfected HEK293T cells with

100-ms depolarizing pulses between -100 mV and 60 mV in steps of 5 mV from a holding potential of -150 mV. For measuring G/V curve, currents were measured with 100-ms depolarizing pulses between -140 mV and -20 mV in steps of 5 mV from a holding potential of -150 mV. For measuring steady-state inactivation, NavEh transfected HEK293T cells were applied pre-pulse potentials between -140 mV and -20 mV in 5 mV increments for 500-ms followed by a 50-ms test pulse at 0 mV. The Boltzmann fitted data yielded activation $V_{1/2} = -61.5 \pm 2.1$ mV (n=15) and steady-state inactivation $V_{1/2} = -94.4 \pm 2.1$ mV (n=9). Data are mean \pm s.e.m.

5. The authors should present a complete set of ion selectivity data for their NavEH channel expression. What is the reversal potential? Is there calcium dependence? What happens when sodium is replaced with NMDG? Is the current sensitive to TTX and STX?

Reply: We thank Reviewer 1's suggestion. Helliwell et al studies showed that NavEh is a sodium selective channel which is regulated by Ca^{2+} from the extracellular side. Our results confirmed these conclusions. In our experimental conditions, the reversal potential is 58.9 ± 5.1 mV (n=9, **new Fig. 1b**), close to the equilibrium potential of Na^+ (66.7 mV at 20°C). When extracellular sodium was replaced with NMDG, no current was observed (See **FigureA** below). When omitting calcium in the extracellular solution, sodium current was completely abolished (See **FigureB** below). We found that NavEh is insensitive to TTX. Applying of 10 μ M TTX only reduced ~20% of peak current (See **FigureA** below). Structural superposition of the NavEh SF vestibule with that of the Nav1.7-TTX structure (PDB ID: 6j8j) reveals that NavEh lacks important interactions for the affinity binding of TTX (See **FigureC** below). For example, a single key residue Tyr362 to Cys374 substitution in the TTX-insensitive Nav1.5 drops the TTX affinity by ~500-fold (PMID: 10681444; 31866066), this critical Tyr or Phe residue for TTX-binding is absent

in NavEh.

6. The voltage dependence of inactivation curve shown in Fig. 1b appears to have at least two components and should be re-analyzed as such. Does this second component indicate calcium dependence of the current?

Reply: We thank Reviewer 1's suggestion. We have carefully re-examined our data and found that some cells have very large currents (>3-fold larger than average current) at -70 to -80 mV in response to the activation protocol. These cells will generate unstable peak currents under voltage-clamp recordings. By excluding these data, we have re-analyzed the data for the voltage-dependence of inactivation curve in the **new Fig.1b**, yielding $V_{1/2}$ of -94.4 ± 2.1 mV (n=9), which is close to Helliwell et al study ($V_{1/2} = -97.2 \pm 0.6$ mV, n=22).

7. The activation curve (Fig. 1b) starts at -100 mV. The data acquired from -140 to -100 mV should be analyzed and included.

Reply: Corrected as requested. See **new Fig. 1b** above.

8. The $V_{1/2}$ values for activation and inactivation of NavEH are quite negative compared to mammalian sodium channels as well as the channels described by Helliwell et al (Current Biology). How do the authors explain this discrepancy?

Reply: The $V_{1/2}$ values for activation and inactivation of NavEh are -61.5 ± 2.1 mV (n=15) and -94.4 ± 2.1 mV (n=9), respectively, which are close to Helliwell et al (Current Biology) study (Activation $V_{1/2} = -68.3 \pm 0.9$ mV (n=14); Inactivation $V_{1/2} = -97.2 \pm 0.6$ mV (n=22)). The $V_{1/2}$ values of NavEh are indeed more negative than that of mammalian sodium channels, for instance, $V_{1/2}$ values for activation of mammalian sodium channels are usually between -40 mV and -20 mV. This difference is probably because the homotetrameric arrangement of the signal domain NavEh channel which responds to depolarizing stimuli synchronously, while the mammalian sodium channels are comprised of four asymmetric domains which employ an asynchronous activation mechanism (PMID: 26712848). In addition, the numbers of gating charges in the VSD (5 in VSD of NavEh; 4 in VSD_I and VSD_{II}, 5 in VSD_{III} and 6-8 in VSD_{IV} of mammalian sodium channels) and the residues interacting with these gating charges are also different (**new Fig. 3g-j**).

New Fig. 3g-j: VSD comparison between Nav_{Eh} and domain I-IV of Nav_{1.5}(PDB code: 7FBS; gray), respectively.

9. How many independent transfection experiments are represented in Fig. 1b?

Reply: The data for **Fig. 1b** were acquired from 8 independent transfection experiments.

10. Full families of current traces and IV curves should be presented in Fig. 3a, instead of representative traces at -35 mV. In addition, the n numbers for this experiment are very low (n=4-6). Finally, how many independent transfection experiments are represented by these data? This comment also applies to the data in Fig. 4.

Reply: We thank Reviewer 1's suggestion. We presented representative current traces at -35 mV in order to quantitatively show the percentage of remaining current at 10 ms (I_{10ms}) after channel opening compared to peak current (I_{peak}). Full families of current traces would be difficult to tell which current trace was measured at -35 mV. We have increased the n numbers for all the experiments in this study to at least 6. As suggested, we have included two additional NavEh current traces at -65 mV and -10 mV in **new Fig. 4a** for clarity. For the data in **new Fig. 4 and Fig. 5**, at least five independent transfection experiments were carried out for data acquisition.

New Figure 4. N-helix determines fast inactivation of NavEh

a, Functional characteristics of NavEh variants. Representative current traces measured at -65 mV, -35 mV, and -10 mV for NavEh^{WT}, NavEh^{Δ2-13}, NavEh^{Δ2-13}+peptide²⁻¹³ and NavEh^{nE5}, respectively. The ratio of current at 10 ms ($I_{10\text{ms}}$) over peak current (I_{peak}) measured at -35 mV were shown in the right panel. Each dot represents a single-cell recording (n=7-9). Significances were determined using two-sided, unpaired t-test. $P < 0.0001$ (NavEh^{Δ2-13}); $***P < 0.0001$ (NavEh^{Δ2-13}+peptide²⁻¹³); $***P < 0.0001$ (NavEh^{nE5}). Data are mean \pm s.e.m.

11. How many independent transfection experiments are represented by the data shown in Extended Fig. 5?

Reply: The data for **new Supplementary Fig. 4 (old Extended Fig. 5)** were acquired from 8, 6, 7, 5, 5 independent transfection experiments for NavEh^{WT}, NavEh^{Δ518-542}, NavEh^{Δ2-13}, NavEh^{Δ2-13}+peptide²⁻¹³ and NavEh^{nE5}, respectively.

12. The authors should discuss potential mechanisms for the rightward shifts in activation shown in Extended Fig. 5.

Reply: The N-terminus or C-terminus of sodium channels and potassium channels are reported to be important for the regulation of channel gating (PMID: 12560340; 21357734). For instance, removal of the N-terminus domain of mammalian sodium channel abolishes the current (PMID: 22739120); a single Glu1784Lys mutation in the DIV-S6 C-terminal end of Nav1.5 shifts the voltage dependence of activation to depolarized direction by +12.5 mV (PMID: 18451998). The NavEh structure revealed the electrostatic interactions between the positively-charged N-helix and the negatively-charged outer-mouth of the activation gate. We postulate that at resting state the N-helix is pre-docked in the position just outside the gate via these strong electrostatic interactions. Deletion of the N-helix or mutations breaking the electrostatic interactions not only inhibit fast inactivation, but also destabilize the activation gate that potentially cause the shift of the voltage dependence of activation to depolarized direction.

13. Fast inactivation is only partially restored by the 2-13 peptide (Extended Fig. 5), rather than “almost fully restored,” as stated in the text (line 172). The authors should discuss this result and at least provide a potential explanation.

Reply: We thank Reviewer 1's suggestion. The recovery of the fast inactivation by the peptide²⁻¹³ is concentration dependent. At high concentration (200 μM), the fast inactivation was restored by $88.6 \pm 6.1\%$ (n=9) when test pulse was held at -35 mV; at lower concentration of 50 μM and 100 μM , the fast inactivation is less restored by $68.1 \pm 8.7\%$ (n=6) and $78.8 \pm 6.7\%$ (n=7), respectively, when test pulse was held at -35 mV. The concentration dependent inactivation of the N-terminal peptide is in agreement with previous results of potassium channel (PMID: 2122520). We have added a discussion about this result and toned down the statement in the revision as below.

174 Fig. 4b). In addition, we found that the fast inactivation of the NavEh^{Δ2-13} can be partially
175 restored when intracellularly applying a synthetic polypeptide of the N-helix (peptide²⁻¹³)
176 in a concentration-dependent manner (Supplementary Fig. 4b). At high concentration of
177 200 μM, the peptide²⁻¹³ can restore the fast inactivation of ~88.6 ± 6.1% (n=9) when test
178 pulse was held at -35 mV (Fig. 4a). These results strongly indicate that the N-helix is

14. Are NavEH currents modulated by mammalian sodium channel beta subunits?

Reply: We thank Reviewer 1's suggestion. From the Nav_v1.7-beta1-beta2 complex structure, we know that the immunoglobulin domain of beta1 forms extensive interactions with the extracellular loop (ECL) of Nav_v1.7 domain1, and the beta2 is anchored to the ECL of Nav_v1.7 domain2 through a disulfide-bond. A single additional glycosylation site on ECL₁ and lacking of the cysteine for disulfide-bond formation in Nav_v1.5 block the binding of beta1 and beta2 to Nav_v1.5, respectively (PMID: 31866066). We performed structural comparison of the NavEh and human Nav_v1.7-beta1-beta2 complex (See **Figure** below). The ECL of NavEh directly clashes with the beta1 subunit, and the cysteine for linking beta2 is absent in NavEh, suggesting that the human beta1 and beta2 subunits may not bind to NavEh.

Structure superposition of pore module NavEh and human Nav_v1.7-beta1-beta2 (PDB code: 6j8j).

Panel A: Superposition of NavEh pore module with domain I PM of Nav_v1.7-beta1-beta2. The ECL of NavEh directly clashes with the beta1. **Panel B:** Superposition of NavEh pore module with domain I PM of Nav_v1.7-beta1-beta2. The cysteine for linking beta2 is substituted by Ser241 in NavEh.

Reviewer #2 (Remarks to the Author):

A manuscript by Jiangtao Zhang et al. describes the cryo-EM structure of NavEh from the coccolithophore *Emiliania huxleyi*. NavEh structure reveals a novel molecular gating mechanism of Nav channel fast inactivation that is similar to the N-terminus mediated inactivation in Kv channels. Experimental testing confirmed key observations about the molecular mechanism of NavEh channel inactivation from the NavEh structure. This study provides valuable structural insights into NavEh channel inactivation compared to classical fast inactivation of eukaryotic Nav channels. The manuscript is very well written, and the data are presented clearly. I have only one minor comment:

Reply: We appreciate Reviewer 2's positive comments on the novelty and significance of this work, and we thank Reviewer 2 for his/her suggestions that helped to improve our manuscript.

1. Move Extended Data Fig. 4 as a new Figure in the main text – Figure 5.

Reply: We have revised and moved the **Extended Data Fig.4** to the main text as **New Fig. 3** in the revision.

Reviewer #3 (Remarks to the Author):

Voltage-gated ion channels play fundamental roles in many biological processes. They are present in all kingdoms of life, from bacteria to human, and have diverse inactivation mechanism. In this manuscript, Zhang et al discovered that NaVEh has an IFM-independent fast inactivation mechanism. High resolution structure of NaVEh and its structure without N-helix reveal that the N-helix plugs into the open pore and blocks the currents. Electrophysiological experiments confirmed the functional role of N-helix in fast inactivation. The work is well carried out and the manuscript is well-written. I have no major concerns but a few minor suggestions for the authors.

Reply: We appreciate Reviewer 3's positive comments on the novelty and significance of this study, and his/her suggestions for improving our manuscript.

1. The authors might use tools such as BLAST to identify if similar N-terminal sequences exist in other ion channels. This might broaden the scope of the current research.

Reply: We thank Reviewer 1's suggestion. We had performed NCBI protein BLAST of the N-terminal sequence. However, we did not find a similar N-terminal sequence in other ion channels. When checking the sequence alignment of the N-terminal sequences that mediate the N-type inactivation in Na_vEh or potassium channels, we found that they share a similar overall pattern of hydrophobic or small side-chain residues followed by a few positively charged residues, but these sequences are variable.

New Supplementary Figure 5a: Sequence alignment of the

N-terminus responsible for N-type inactivation from Na_vEh

(Accession number CAMPEP_01 87645740), Na_vSa1

(Accession number CAMPEP_01 19314838), Na_vSa2

(Accession number CAMPEP_01 19345692), MthK

(accession number CEP 36137), Shaker B (accession

number CAA 29917), Kv-β1.1 (accession number CAA

50000), BK-β3a (accession number NP_741979) and BK-β2a (accession number NP_001265840).

a

		Arg-cluster	
NavEh	1	MIAAIHNA RR-KKR EAAAAHK	20
NavSa1	1	MVVASSTLGS-QAAHAHYRAN	20
NavSa2	1	MYLRITNIVES-SFFT K FIIY	20
MthK	1	MVLVIEI IRK -HLP RVLK VPA	20
ShakerB	1	MAAVAGLYGL-GED RQHRKKQ	20
Kv-β1.1	1	MQVSI ACTEH -NL KSR NGED R	20
BK-β3a	1	MQPFSIPVQI-TLQGS RRR Q G	20
BK-β2a	1	MFIWTS GR TS-SSY RHDEKRN	20

2. NaVEh is not a well-characterized ion channel. The authors claimed that there is EF hand domain in C terminus

and the 2nd 3D class seems to have stronger densities for such helices. Would the resolution of the 2nd 3D class improved to a level where EF hand domain could be modeled? More importantly, are the channel behaviors modulated by calcium?

Reply: We thank Reviewer 3 for raising this important point. We agree with Reviewer 3 that the novel channel NavEh, which was identified in 2020, is a less well-characterized channel in terms of regulation by Ca²⁺ or the intracellular EF-hand like domain. Sequence alignment shows that the C-terminal EF-hand like domain of NavEh is conserved with the canonical EF-hand proteins (See **FigureA** below). In addition, the AlphaFold2 model of the NavEh EF-hand like domain can be well superimposed with EF-hand protein of human PrvB (See **FigureB** below). We also re-calculated the 3D classes to generated an improved EM map for the C-terminus, which could accommodate most helices of the EH-hand like domain of the AlphaFold2 model (See **FigureC** below). Furthermore, like NavEh, human Nav channels also possess EF-hand like domains at the C-terminus, and the crystal structures of Nav1.5 C-terminus domain revealed that its EF-hand like domain shares similar folding to canonical EF-hand protein structure (PMID: 19074138; 22705208; 31072926).

We also tested the modulation of NavEh by Ca²⁺ and found that the Na⁺ current of NavEh is regulated by extracellular Ca²⁺. Omitting the extracellular Ca²⁺ completely abolishes the sodium current (See **Figure right**), which is surprising but consistent with Helliwell's study (PMID: 33004614). We speculate that there might be unidentified Ca²⁺ binding sites in the extracellular side of NavEh, or Ca²⁺ stabilizes the SF of NavEh. The potential mechanisms for Ca²⁺ regulation need further investigation.

3. The binding of small N-helix to the 4-fold axis complicated the computation and map interpretation. Refinement with C1 symmetry would be mainly driven by the signal of channel. Therefore, is it possible to use symmetry expansion and focused classification with high “Tau numbers” but without alignment to separate the four conformations of N-helix?

Reply: We thank Reviewer 3's suggestion. We performed symmetry expansion of the final particles with C4 symmetry, then performed skip-alignment 3D classification with “Tau=20” and a mask only covering one protomer (See **FigureA** below). The resulting 3D classification shows that the positions of the N-helix are consistent, no obvious conformational changes were observed (See **FigureB** below). The C-terminal end of the N-helix is slightly tilted to the VSD, indicating its flexible connection to the VSD.

Reviewers' Comments:

Reviewer #1:

Remarks to the Author:

The authors have addressed all of my critiques. I have no further concerns.

Reviewer #3:

Remarks to the Author:

The authors have addressed all of my previous questions in the revised manuscript. I have no more questions.